# A Structural Framework for Assessing the Digital Resilience of Enterprises in the Context of the Technological Revolution 4.0

Anca Mehedintu *[ID] and Georgeta Soava [ID]

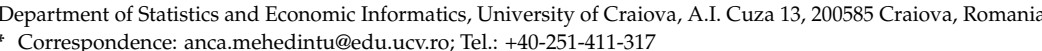

Department of Statistics and Economic Informatics, University of Craiova, A.I. Cuza 13, 200585 Craiova, Romania
* Correspondence: anca.mehedintu@edu.ucv.ro; Tel.: +40-251-411-317

**Abstract:** This research aims to develop a conceptual model to establish the influence of digital core investment and digital innovation on digital resilience at the enterprise level. The data were collected through a questionnaire-based survey of managers and IT specialists of companies. The analysis was performed using structural equation modeling with SPSS Statistics and Amos software. Based on the literature review, the study identifies the main factors that can ensure digital resilience and assesses their impact on Romania's private and public companies. The research results confirm the hypotheses presented in the article, emphasizing that digital resilience is the result of the collaboration of several factors with different effects, determined by using Industry 4.0 technologies. Thus, digital core and digital innovation investments help improve digital resilience. Moreover, digital core investments have a positive impact on the digital resilience of enterprises, mediated by digital innovation investments. The study's novelty consists in the realization of a model of interconnected analysis of several variables specific to digital and innovative technologies to ensure the resilience framework at the company level. The research offers valuable results which can be used by companies in Romania or other European Union countries to ensure their digital resilience.

**Keywords:** technological revolution 4.0; digital transformation technologies; cloud computing; Internet of Things; digital innovation; digital resilience; structural equations modeling

## 1. Introduction

In a global, interconnected world, companies, regardless of their size, understand that to develop they must adopt new innovative technologies and introduce Industry 4.0 technologies throughout the business process. At the same time, the COVID-19 crisis has shown that systemic crises are more likely than ever, so companies need to be prepared for such shocks. The COVID-19 crisis has demonstrated to company managers the central role of digital technologies, both in keeping their operations running and innovating and pivoting to new requirements. The effects of the pandemic crisis, further exacerbated by the outbreak of the Russia–Ukraine war, make business resilience more important than ever, making it practically a necessity for companies to remain competitive and to maintain their level of confidence in the market.

The importance of resilience for global regulators is not new, with the first significant measures being taken following the 11 September 2011 terrorist attacks, and the 2008 financial crisis strengthening them. However, the COVID-19 crisis has caused a high level of disruption and business risk worldwide, for which no one was prepared.

In the current context of the crisis caused by COVID-19, increasing investment in digital technologies is a challenge for all states. To this end, the European Commission has set up a Recovery and Resilience Mechanism to mitigate the economic and social impact of the pandemic and to increase the sustainability and resilience of European economies to better prepare them for the challenges and opportunities posed by the green and digital transitions. The recovery plan for Europe adds resilience as a key dimension of European Union (EU) progress to significantly advance on the digital transition. Thus, EU Member

States are allocating more than 26% (26.4%) of national recovery plans to digitization expenditure, which far exceeds the value of the initially forecast expenditure for 2050 (20%) [1]. The lines of action of this mechanism offer the possibility for each EU state to allocate funds for the adoption of technologies, such as artificial intelligence, Internet of Things (IoT) connectivity, and cloud computing, to ensure digital resilience [2] while also identifying best practices, which support companies in adopting digital technologies [3].

At the Romania level, the application of the Recovery and Resilience Mechanism is achieved through the National Recovery and Resilience Plan [4] to carry out programs and projects to support resilience, the level of preparedness for crises, adaptability capacity, and growth potential. By creating a coherent digital infrastructure, Romania will be able to increase the quality of digital services for both citizens and companies, so as to increase the performance of Romanian companies in terms of digitalization and sustainability compared to the EU benchmark [5].

The technological revolution 4.0 and the COVID-19 pandemic marked a turning point in the digital transformation, so the business strategy has changed, emphasizing the use of digital capabilities and rapid adaptation to capitalize on the new conditions and maintain business continuity [6,7].

All states, regardless of their level of development, have in mind the importance of redesigning strategies for resilient and sustainable regional economic development [8], improving regional resilience [9], and minimizing the cost of recovery as fundamental to development [10].

In the face of globalization, the COVID-19 crisis has had a significant impact on the disruption of the global economic sector, including for start-ups, forcing entrepreneurs to pursue a continuous process of innovation to secure their future business by ensuring a business resilience framework [11]. Large companies which have digital platforms have found ways to thrive amid this crisis [12], as opposed to small and medium-sized enterprises (SMEs), which have proven to be vulnerable to shocks, causing significant damage to them [13–15]. Thus, researchers have highlighted the importance and role of entrepreneurship in rebuilding local economies and preparing their resilience for critical times [16,17], as well as the opportunities and challenges inherent at the intersection of disaster recovery and building resilience [18]. In this sense, small businesses can increase their resistance to natural hazards while contributing to ensuring community resilience [19].

The digital society allows the assertion of entrepreneurial ecosystems and the integration of entrepreneurial and technology-related dimensions into a single unified model [20].

Several studies have explored the main factors of innovation and business success in different organizations and the environments in which they operate [21] to achieve effective resilience management [22].

The resilience and agility of digital infrastructure have been shown to be essential in the times of uncertainty and disruption that organizations face [18]. During the COVID-19 pandemic, several IT-related challenges arose from the perspective of both managers and IT professionals [23]. Based on the contradictions in the digital transformation process of companies, it is very important to identify whether companies' investments in digitalization can build organizational resilience [24]. Use of the existing digital technology may not be sufficient to cope with the COVID-19 crisis, so it must be accompanied by a digital reorganization [25] or a comprehensive business reorganization [26].

Following the research of the literature on building digital resilience in organizations, it was found that there is little information on this issue. In this sense, this paper's purpose is to evaluate the digital resilience model of Romanian companies by examining the most important components of the digital core investments and innovation investments and their causal effects. The motivation for this study stems from the desire to know how company managers are aware of the importance of investing in digital technologies and integrating digital innovations in building digital resilience following the devastating impact of the COVID-19 crisis on their business.

Starting from the digital resiliency investment index established by the International Data Corporation [27] study, a conceptual model is developed and proposed. Through the modeling of structural equations (SEM), a series of hypotheses are tested and the proposed model is validated. It is found through this research that investing in digital technologies and innovation helps businesses improve their digital resilience. It is also found that core digital investment and investment in innovation have a significant positive impact on the digital resilience of Romanian companies.

The benefit of this study is primarily for company managers who can understand the need to adopt a strategy based on digital transformation and innovation to build digital resilience. It can also be the basis for other research studies and can provide management suggestions and policies for the management practices of EU companies.

The paper is structured in five sections. Section 2 discusses the theoretical background and identifies the hypotheses on which the research model is developed. Section 3 describes the sample and presents the methodology used for modeling structural equations. Section 4 presents the results obtained, and Section 5 summarizes the main contributions of this study, the theoretical and managerial implications, and highlights the main limitations of the research and some possible future research directions.

## 2. Theoretical Background and Hypotheses

### 2.1. Industry 4.0 Technologies, COVID-19, and Digital Resilience

The resilience of a business is the ability of a company to respond, recover, and resume operations at a level of service acceptable to consumers, customers, and contractual partners in the event of significant disruptions [28].

A review of the literature on Industry 4.0 technologies, crisis management, and resilience [29,30] emphasizes the importance of the digital transformation of enterprises [31] to manage crises that may threaten the organization's survival [32,33].

Several studies have confirmed that digital transformation is essential for the modern economy, bringing several benefits to companies depending on the digital capacity adopted [34]. In this context, Industry 4.0 technologies are a fundamental tool for economic recovery [3].

However, the use of digital technology can also have some negative effects due to the social distancing requirements imposed during the COVID-19 crisis. For example, human labor has been reduced by partially replacing human labor with machines [35], and cybersecurity issues have increased through remote work and the use of online platforms [36]. The coronavirus pandemic has transposed many businesses online, so the companies located in areas with poor access to digital connectivity have limited capacity to develop resilience in difficult economic times [37].

In this context, it is obvious that digital technology supports the process of business transformation, and Industry 4.0 technologies provide the necessary impetus for the long-term development of society as a whole, increasing resilience to crises and adapting the workforce to new market conditions to be competitive. The International Data Corporation (IDC) has established a digital resilience investment index that allows companies to track the extent to which their technology investments, customers, competitors, and business partners are moving towards building the concept of digital resilience [27]. The index provides a composite picture of the digital resilience of companies with two elements: digital core investment and digital innovation investment.

#### 2.1.1. Security Investments

Information is becoming increasingly important for individuals and organizations, and ensuring their security is a major concern of business managers who need to make optimal investments in ensuring information security and customer confidentiality [38]. Information sharing has become a significant part of companies' security efforts, and the optimal level of sharing can be conditioned by the budget allocated to protection and security [39,40]. The implementation of security applications protects against threats, but

they also tend to access a range of information that invades privacy [41]. Organizations that invest in security systems also consider critical infrastructure security, as this refers to the protection of systems, networks, and assets whose operation ensures statewide security and, if computing resources are limited, it is impossible to run anti-malware programs [42].

The rapid development of information technology in the network, the Internet of Things, and devices in industrial environments increasingly connected to the Internet lead to a permanent exchange of data that raises several security issues [43]. This requires strengthening the enterprise network security so as to ensure the network information security [44]. At the same time, security is a concern for companies moving to the cloud to provide technologies and services that protect data and applications in the cloud from threats. Cloud security is the shared responsibility of the cloud provider (protecting the infrastructure itself, as well as accessing, fixing, and configuring the physical hosts and physical network on which the computers run, as well as storage and other resources) and cloud users (managing users and their access privileges, protecting cloud accounts from unauthorized access, encrypting and protecting cloud-based data assets, and managing its security position) [45].

### 2.1.2. Remote Working

Although distance work has captured the interest of managers since the implementation of new information and communication technologies (ICT), the COVID-19 pandemic and the interim period accelerated the change in their attitude towards distance work, which became a necessity for many organizations [46], with all the challenges posed by the new type of work [47]; employment may be fully remote (working full time from their locations for a company that has a traditional office), flexible (offers some flexibility with schedule, location, or both), or may not allow remote work. Company managers are increasingly focusing on the consequences of remote work [48] and on the factors that facilitate its effectiveness in the post-COVID-19 era [49] from the perspective of employees and employers [50].

### 2.1.3. Investments in Cloud Migration Services

Organizations, regardless of size, driven by the need for higher productivity and lower costs, are moving their application portfolio to the cloud. In addition to the many benefits of migrating applications to the cloud, there are several critical challenges that companies face while adopting the cloud [51,52], which have different implications depending on the size of the organizations [53], the variety of cloud providers [54], and whether managers choose evolutionary or transformative digital migration [55]. Thus, an assessment of cloud running costs and accessibility of migration [56], as well as security issues (threats and attacks that may occur), is needed to take appropriate countermeasures.

Among the cloud service models, the software as a service model (SaaS) has been an option chosen by many organizations [57] because they are able to transfer some or all of their IT features to a cloud service provider. Other companies migrate their databases to various complete, flexible, and cost-effective cloud platforms for developing, running, and managing applications—platform as a service (PaaS) [58]. One form of cloud computing that provides organizations with basic computing, networking, and on-demand storage resources over the Internet and with pay-as-you-go service is "IaaS" (infrastructure as a service), which allows users to expand or reduce resources as needed [59]. Organizations can also opt for a hybrid cloud when the demand for computing and processing grows beyond the capabilities of a local data center (private cloud), or companies can use a public cloud, allowing them to avoid the time and cost of purchasing, installing, and maintaining new servers they do not always need [60].

### 2.1.4. Digital Transformation Investments

Business disruptions have substantial negative consequences on sales profitability, profit and inventory performance, brand image, employment, buyer safety, and overall supply performance [61]. This has led to the need to use digital technologies to modify or create new business processes to implement innovative techniques and behaviors [62] based on culture and the experience of the customers to meet the requirements of an ever-changing market. Through the Internet and the implementation of websites, organizations can make their brand known through sustained online and offline business promotion, generating contests and promoting events, conducting direct sales, and discovering community members interested in their products [63,64]. At the same time, social networks play an important role in the digital transformation of business. Companies using the right tools can significantly increase business efficiency [65]. By using chatbot interfaces built into any messaging app, organizations can interact flexibly with customers (as a human would) at any time of the day or week and are not limited by time or physical location, so the costs are negligible [66].

Mobile applications have grown in popularity in recent years, and organization managers have identified their usefulness and ease of use in business processes [67], significantly influencing the behavior of their users [68].

### 2.1.5. Digital Adaptation

Digital technologies facilitate general changes in business models, with companies being able to adopt multiple business models to serve different market segments [69]. Several digital management tools have been developed to stimulate innovation in the current age of digital transformation [70]. In the context of the technological revolution 4.0, it has become necessary for company managers to establish a digital adaptation strategy [71], to use technological innovation to implement robust and sustainable strategies [72–74], and to try to return to a new normal [75,76]. The activity of companies has intensified remote work and the use of private devices, digital tools, ICT software, and online platforms for work, reflecting profound changes in business processes and models and demonstrating that employees have become better integrated into digital environments.

### 2.1.6. Digital Acceleration

As digital technologies penetrate and integrate into all walks of life, organizations are facing increasing pressure to apply digital innovation in an accelerated way to update and transform their business models to remain competitive. At the same time, the restrictions introduced by the COVID-19 pandemic have forced organizations to adopt a technology-intensive business model that adapts quickly to the disruptive environment [77]. It has also produced several changes in the approach to business relations as an effect of the accelerated digital transformation processes driven by the pandemic crisis [78]. At the same time, it has been shown that employees adopt digital innovations faster if they are directly involved in this process [79]. There is a significant connection between digital organizational culture and digital capabilities with digital innovation, and organizational training can be a mediator between digital capabilities, digital organizational culture, and digital innovation [80]. There are several new investments in technology and business models through digital acceleration so that digital customers are involved as effectively as possible.

### 2.1.7. Digital Core

Managers prioritizing business digitization must consider the following main factors: accessibility, availability, awareness, and acceptability [81]. An organization's digital core includes the platforms and technological applications that allow it to meet the needs of the digital economy. It may include state-of-the-art technologies: cloud computing, security, advanced data analysis, the Internet of Things, artificial intelligence, collaborative support for remote workers, and digital transformation projects. The COVID-19 pandemic has

forced businesses to transform at an unprecedented rate, and the rapid deployment of digital technologies is of paramount importance [21] because the operational efficiency of a business depends on its responsiveness, scalability, and digital infrastructure resilience [82]. Moreover, the attitude of companies towards remote work has changed in a positive sense [46], the flexibility of remote work being facilitated by organizational support [83]. The adoption of Industry 4.0 at the company level can support business management [84,85] in the fight for survival and recovery in the COVID-19 era. It offers the opportunity to redesign and automate production processes and adopt new business models [86], improving productivity [87] and accelerating the production process [3], which has a direct influence on the company's performance [88].

### 2.1.8. Digital Innovation

Investments in digital innovation take into account current and projected investments in IT, including the volume of expenditure (new or reallocated) on digital resilience (DR). In the face of globalization, the technological revolution 4.0 and the COVID-19 pandemic have forced organizations to pay more attention to the innovation process to create agility and continuous adaptation [72] to ensure the future of their business [11]. Companies approach digital innovation in the context of rethinking and transforming existing products [89] and allocate a significant portion of the budget to innovation processes [62,82] to achieve a digital resilience framework [22]. In many cases, digital resilience transformation and creation have been achieved through the adoption of Industry 4.0 technologies [3]. At the production level [82,90,91], it allows production systems to identify and manage anomalies and disruptive events and to support decisions to mitigate their consequences [92]. At the level of the supply chain [93–95], it achieves efficient, resilient, and sustainable supply chains [96,97], having the possibility to use criteria for selecting and prioritizing suppliers [98]. Resilience allows supply chains to reduce their predisposition to disruptions and recover faster [99]. The adoption of Industry 4.0 offers the possibility for supply chain managers to define its roadmap [100], and interoperability can create ways to ensure post-COVID-19 resilience [101].

### 2.1.9. Digital Resilience

In a digital world marked by complexity and uncertainty, it is clear that resilience is essential for companies not only to survive, but also to thrive [102]. The interaction between management and digital technologies facilitates a company's performance and the achievement of organizational resilience [103]. Digital platforms played decisive roles during the COVID-19 crisis and facilitated the transition from recovery resilience to transformative resilience [104]. The concept of the resilience of the digital platform ecosystem has been developed to cope with exogenous shocks and become resistant to future disruptions [105].

Digital transformation may be effective in recovering from the COVID-19 pandemic [106], but can also cause several disruptions in a company in response to strategies used in managing structural change caused by the implementation of digital technologies [107]. Thus, organizations managers, to build the company's digital resilience, must develop a business model based on organizational culture [80] so that the allocation of investments can be directed towards digital innovation [103], which can be supported by employees as well [50].

### 2.2. Description of Research Hypotheses and Structural Model

In the theoretical context of structural equation modeling, we integrate the concepts of digital resilience and propose an extended structural model that identifies the input variables (predictors) and allows the analysis of their influence on the output variable (digital resilience). To achieve economic modeling and the research objectives, we propose a conceptual model and formulate the following hypotheses:

**Hypothesis 1 (H1).** *Security investments (SI) have a positive and significant effect on digital core (DO).*

**Hypothesis 2 (H2).** *Remote working (RW) has a positive and significant effect on digital core (DO).*

**Hypothesis 3 (H3).** *Cloud migration (CM) has a positive and significant effect on digital core (DO).*

**Hypothesis 4 (H4).** *Digital transformation investments (DX) have a positive and significant effect on digital core (DO).*

**Hypothesis 5 (H5).** *Cloud migration (CM) has a positive and significant effect on digital innovation (DI).*

**Hypothesis 6 (H6).** *Digital transformation investments (DX) have a positive and significant effect on digital innovation (DI).*

**Hypothesis 7 (H7).** *Digital adaptation (DA) has a positive and significant effect on digital innovation (DI).*

**Hypothesis 8 (H8).** *Digital acceleration (DC) has a positive and significant effect on digital innovation (DI).*

**Hypothesis 9 (H9).** *Digital core (DO) has a positive and significant effect on digital innovation (DI).*

**Hypothesis 10 (H10).** *Digital core (DO) has a positive and significant effect on digital resilience (DR).*

**Hypothesis 11 (H11).** *Digital innovation (DO) has a positive and significant effect on digital resilience (DR).*

Integrating the relations presented in the previous hypotheses and postulating the mediation effect, we formulated Hypothesis 12.

**Hypothesis 12 (H12).** *Digital core (DO) mediates the relationship between security investments (SI), remote working (RW), cloud migration (CM), digital transformation investments (DX), and digital innovation (DI), and digital innovation (DI) mediates the relationship between digital core (DO) and digital resilience (DR).*

Starting from the two components, digital core investments and digital innovation investments [27], nine constructs were considered in this study: (1) security investments (SI); (2) remote working (RW); (3) cloud migration (CM); (4) digital transformation investments (DX); (5) digital adaptation (DA); (6) digital acceleration (DC); (7) digital core investments (DO); (8) digital innovation investments (DI); and (9) digital resilience (DR) (Table 1).

**Table 1.** Constructs, items, and coding for structural model.

| Latent Variables | Construct | Items | Coding | Source |
|---|---|---|---|---|
| Exogenous Latent Variables | Security investments (SI) | The company ensures Internet of Things security | SI1 | [42] |
| | | The company uses solutions application security | SI2 | [41] |
| | | The company ensures network security | SI3 | [44] |
| | | The company uses solutions cloud security | SI4 | [45] |
| | | The company uses solutions for critical infrastructure security | SI5 | [43] |
| | Remote working (RW) | The company allows fully remote work | RW1 | |
| | | The company allows hybrid remote work | RW2 | [46–50,83] |
| | | The company does not allow remote work | RW3 | |
| | Cloud migration (CM) | The company uses software as a service (SaaS) | CM1 | [57] |
| | | The company uses platform as a service (PaaS) | CM2 | [58] |
| | | The company uses infrastructure as a service (IaaS) | CM3 | [59] |
| | | The company uses hybrid model | CM4 | [60] |
| | Digital transformation investments (DX) | The company uses tools for interaction through the website, e-commerce, and m-commerce | DX1 | [64] |
| | | The company uses mobile applications in business processes—Apps | DX2 | [67,68] |
| | | The company uses social networks in the digital transformation of business—Social media | DX3 | [65] |
| | | The company uses conversational interfaces—Chatbots | DX4 | [66] |
| | Digital adaptation (DA) | The company implements IT projects to support vulnerable processes discovered in times of crisis | DA1 | [69,71–76] |
| | | The company develops IT projects in support of the new operational requirements generated by the implementation of technology 4.0/pandemic crisis | DA2 | |
| | Digital acceleration (DC) | The company develops IT projects that model business innovation | DC1 | [70,77–80] |
| | | The company develops IT projects to increase market share | DC2 | |
| Endogenous Latent Variables | Digital core (DO) | The company uses virtual individual work | DO1 | [46,83] |
| | | The company adopts industry 4.0 technologies in the automation process | DO2 | [3,55] |
| | | To what extent has the company achieved digitizing the business | DO3 | [24] |
| | | To what extent the digitalization of the business has led to its globalization | DO4 | [8] |
| | Digital innovation (DI) | The company introduces digital products | DI1 | [3,11,22,62, 72,82,84,89] |
| | | The company uses customer touch points and gives enhanced sales pitches | DI2 | |
| | Digital resilience (DR) | The company has established a strategy for developing the digital business model | DR1 | [50,51,69,80, 102–107] |
| | | The company has improved customer experience: customer journeys, channels, and touchpoints | DR2 | |
| | | The company uses platforms and infrastructure for digital processing of the data and information | DR3 | |

Following the revised literature and the hypotheses formulated, we propose a conceptual model in which the links between latent variables are presented in Figure 1.

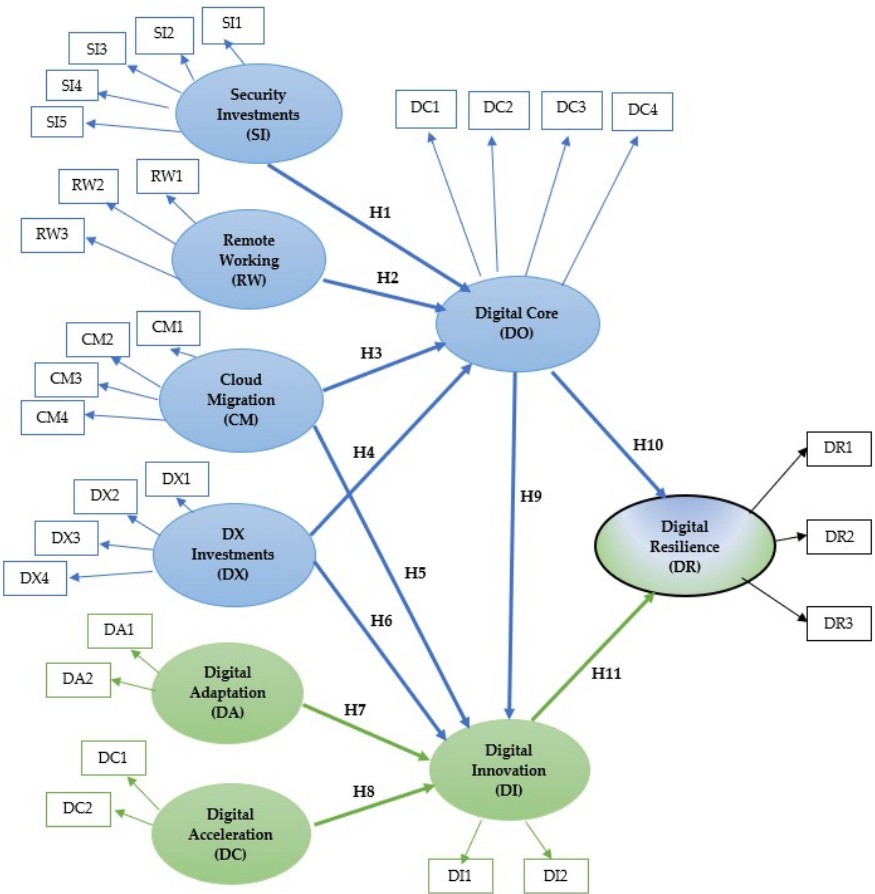

**Figure 1.** Description of the conceptual structural model. Source: Developed by authors.

## 3. Methodology

### 3.1. Data Collection and Sample Characteristics

To analyze the extent to which companies' investments in digital technology led to them building digital resilience, a quantitative survey was conducted based on a 16-question questionnaire. The questionnaire was addressed to IT managers and specialists from Romanian companies and public institutions. For sampling, we used a stratified survey. We considered all types of companies (small, medium, and large) and public institutions, and then we extracted a random sample from each layer. In the sample were selected managers and IT specialists from all categories of companies and public institutions to have a broader perspective on organizational performance in terms of building digital resilience. The selected companies are among the most important in Romania, and the respondents were selected by a non-probabilistic sampling method. The questionnaire was distributed both offline, during business meetings on the occasion of the awarding of the best managers (April 2022) and during personal meetings and phone calls, and online, via e-mail, WhatsApp, and several social networks (LinkedIn, Twitter, and Facebook). The data collection took place between 1 September 2021 and 10 June 2022.

From the 16 questions of the survey (Appendix A, Table A1), 9 questions were identified following a review of the digital transformation literature to ensure digital resilience (Table 1); 4 questions were about company characteristics (related to company type and size, business environment, age, and share of revenue allocated to digitization) (Table 2); and 3 questions were to determine the respondent characteristics (gender, age, and education) (Table 2). Thus, the first part of the questionnaire includes the characteristics of the respondents and companies that were the subjects of the analysis (Table 2). The second part includes the measurement elements specific to the items and latent variables related to the proposed structural model that were converted into content questions (Table 1).

**Table 2.** Sample structure.

| Variables | Category | Number (N) | Percentage (%) |
|---|---|---|---|
| | Managerial features | | |
| Gender | Female | 143 | 32.80 |
| | Male | 293 | 67.20 |
| Age | <25 | 32 | 7.34 |
| | 25–30 | 67 | 15.37 |
| | 31–40 | 138 | 31.65 |
| | 41–50 | 124 | 28.44 |
| | >50 | 75 | 17.20 |
| Education | College diploma | 35 | 8.03 |
| | Bachelor's degree | 151 | 34.63 |
| | Master's degree | 212 | 48.62 |
| | Ph.D. degree and above | 38 | 8.72 |
| | Company features | | |
| Type of enterprise | Small/Private enterprise | 159 | 36.47 |
| | Medium/Private enterprise | 147 | 33.72 |
| | Large/Private enterprise | 59 | 13.53 |
| | Public/State-owned enterprise | 71 | 16.28 |
| IT budgets (IT) | <5% of total revenue | 137 | 31.43 |
| | 5–10% of total revenue | 165 | 37.84 |
| | 11–20% of total revenue | 86 | 19.72 |
| | >20% of total revenue | 48 | 11.01 |
| Enterprise's age | <5 | 110 | 25.23 |
| | 5–10 | 94 | 21.56 |
| | 11–20 | 125 | 28.67 |
| | >20 | 107 | 24.54 |
| Enterprise location | Urban | 320 | 73.39 |
| | Rural | 116 | 26.61 |
| | Total | 436 | 100 |

Source: Survey data processing by the authors.

Responses for all 16 questions were collected using a Likert scale that provides response categories on a scale of 1 to 5, where response options included "not at all = 1", "very little = 2", "medium = 3", "much = 4", and "very much = 5" [108]. All respondents were previously contacted and invited to participate in the research. The final form of the questionnaire is the result of a pre-test with potential research subjects which was conducted several times until all the problematic elements had been reviewed and eliminated [109], which contributed to its improvement in terms of the content and structure.

Out of the 950 questionnaires distributed to the respondents, 512 questionnaires were collected, of which 436 were valid, and the remaining 76 were incomplete or incorrectly completed, resulting in a response rate of 45.89%. The margin of error corresponding to a 95% probability of guaranteeing the research results is ±3.17%, which proves that the sample is representative (margin of error < 5%) [110].

In terms of gender, 67.20% of the respondents (managers and IT specialists) were men and 32.80% were women. More than half of the IT managers and specialists (60.09%) fell into the age group of between 31 and 50 years, and 91.97% graduated from university. Regarding the share of revenues allocated for investments in digital technologies, 31.43% of companies allocated a percentage of up to 5%, 37.84% between 5 and 10%, 19.72% between 11 and 20%, and 11.01% over 20%.

The classification of enterprises was made according to the number of employees and the net turnover by the methodology applied by the European Commission [111].

### 3.2. Structural Equation Modeling (SEM)

Digital resilience measurement indicators and their drivers all have abstract and multidimensional characteristics. The variables involved have subjective characteristics, and are difficult to measure directly, producing large measurement errors and complicated causality. Variables contain multiple observable indicators that can be correlated. In this sense, the modeling of structural equations was used for the analysis of this complex system, the testing of research hypotheses, and the analysis of the relationships between variables similar to other research [32,72,83]. SEM is an intense tool used for multivariate economic analysis that allows the examination of a set of relationships between one or more independent variables and one or more dependent variables [69,72]. By using the multivariate linear statistical modeling method SEM, the simultaneous processing of several associated dependent variables, the simultaneous estimation of the structure and the relation of the factors, and the estimation of the degree of fit of the model as a whole can be performed [112,113]. Offering the possibility of multiple regression analysis and confirmatory factor analysis to analyze the relationship on several levels, we consider SEM to be a beneficial exploration to study digital resilience [112,113].

In the SEM model, we first consider whether the model produces an estimated population covariance matrix consistent with the sampled (observed) covariance matrix. Then, the adequacy of the model (through the statistics of the chi-square test and goodness-of-fit indices), the reliability of the indicators, and the estimation of the parameters for each path in the model are taken into account (pathways in predicting the outcome measure, i.e., an independent variable affects a specific dependent variable) and mediation or indirect effects testing (the independent variable affects the dependent variable through a mediation variable) is conducted [112–115].

The SEM method sets the metric model equation firstly:

$$x = \Lambda_x \xi + \delta \tag{1}$$

$$y = \Lambda_y \eta + \varepsilon \tag{2}$$

Equations (1) and (2) stipulate the relationship between the result latent variable $\eta$ and the result observable variable $y$, and the relationship between the cause latent variable $\xi$ and the cause observable variable $x$. $\Lambda_x$ is the relationship between the cause observable variable and the cause latent variable, and is the factor loading matrix of the cause observable variable on the cause latent variable. $\Lambda_y$ is the relationship between the result observable variable and the result latent variable, and is the factor loading matrix of the result observable variable on the result latent variable; $\delta$ is the error of the cause observable variable $x$; and $\varepsilon$ is the error of the result observable variable $y$. The structural model equation is set as follows:

$$\eta = \beta \eta + \Gamma \xi + \zeta, \tag{3}$$

where $\beta$ is the coefficient matrix of the result latent variable $\eta$ and the path coefficient matrix between the result latent variables; $\Gamma$ is the coefficient matrix of the cause latent variable $\xi$ and the path coefficient matrix of the cause latent variable to the corresponding endogenous latent variable; and $\zeta$ is the residual term of the structural equation, which is the part that is failed to be explained within the model.

The collected data were processed using IBM SPSS Statistics 26.0 software [116] to verify the correctness of the information provided by the respondents, eliminate errors, and ensure the interactive validation of the data provided and the implicitly of the questionnaire. Research hypothesis testing and structural model validation were performed using IBM SPSS Amos 26.0 software [116].

## 4. Results and Discussion

### 4.1. Reliability and Validity Model

Before the actual analysis of the model, the degree of significance of the variables of the structural conceptual model was verified and the reliability, validity, and internal consistency of the collected data were analyzed, respectively (Table 3).

**Table 3.** The measurement model results.

| Construct | CA | CR | DG rho | AVE | SR AVE | VIF |
|---|---|---|---|---|---|---|
| Security investments (SI) | 0.883 | 0.889 | 0.904 | 0.667 | 0.817 | 1.007 |
| Remote working (RW) | 0.861 | 0.865 | 0.993 | 0.683 | 0.827 | 1.304 |
| Cloud migration (CM) | 0.922 | 0.927 | 0.993 | 0.762 | 0.873 | 1.172 |
| Digital transformation investments (DX) | 0.919 | 0.918 | 0.934 | 0.739 | 0.860 | 1.219 |
| Digital adaptation (DA) | 0.862 | 0.871 | 0.935 | 0.772 | 0.879 | 2.237 |
| Digital acceleration (DC) | 0.922 | 0.923 | 0.915 | 0.857 | 0.926 | 1.426 |
| Digital core investments (DO) | 0.892 | 0.900 | 0.905 | 0.693 | 0.833 | 2.086 |
| Digital innovation investments (DI) | 0.824 | 0.831 | 0.945 | 0.711 | 0.843 | 1.562 |
| Digital resilience (DR) | 0.839 | 0.848 | 0.910 | 0.652 | 0.807 | - |

Note: Cronbach's alpha (CA); composite reliability (CR); Dillon–Goldstein's rho (DG rho); average variance extracted (AVE); square root of AVE (SR AVE); variance inflation factor (VIF). Source: Own calculations based on survey data processing.

The stability and consistency of the measured results were mainly determined through reliability. In this study, to check the internal consistency reliability of the questionnaire, Cronbach's alpha, composite reliability, and Dillon–Goldstein's rho were used. They were all found to have values either greater than 0.9 (very high reliability) or between 0.8 and 0.9 (high reliability) [117,118], i.e., the exogenous variables in the model are statistically significant. The results of Table 3 show that there are not multicollinearity problems between the exogenous variables in the model, as the values of the variation inflation factor (VIF) do not exceed the value 5 [119].

Validity is measured by whether the variables in the model can really be expressed by the appropriate measurement elements. The validity of the model is higher as the measurement results are more consistent with the content to be measured. In this study, the Kaiser–Meyer–Olkin (KMO) test and Bartlett's test of sphericity were used for validity analysis. The KMO test is used to compare the relative size of the Pearson correlation coefficient and Pearson correlation coefficient among the original variables. The KMO value is between 0 and 1. The correlation between the variables is stronger and more suitable for factor analysis as the value is closer to 1 [120]. Bartlett's test of sphericity is used to test whether the correlation matrix is an identity matrix, that is, whether each variable is independent, indicating that the data are not suitable for factor analysis. However, the lower the level of significance, the greater the likelihood of a significant relationship between the original variables. The test results of this study are shown in Table 4.

**Table 4.** Kaiser–Meyer–Olkin and Bartlett's test.

| Kaiser–Meyer–Olkin Measure of Sampling Adequacy | | 0.709 |
|---|---|---|
| Bartlett's Test of Sphericity | Approx. Chi-Square | 978.793 |
| | df | 378 |
| | Sig. | 0.000 |

Source: Survey data processing by the authors.

In Table 4, the KMO value is 0.709, indicating that the factor analysis effect is good. The value of Bartlett's test of sphericity is 978.793. When the degree of freedom is 378, $p < 0.05$, reaching the significance level. Therefore, this study is suitable for factor analysis.

The average variance extracted (AVE) indicator was used to test the discriminant validity of the model. Thus, the square root of the AVE in each construct was compared with its inter-construct correlation for all latent variables in the model and was observed to be higher, confirming the discriminant validity of the proposed model (Table 5) [121].

**Table 5.** Fornell–Larcker criterion analysis for discriminant validity.

| | SI | RW | CM | DX | DA | DC | DO | DI | DR |
|---|---|---|---|---|---|---|---|---|---|
| SI | 0.817 | | | | | | | | |
| RW | 0.180 | 0.827 | | | | | | | |
| CM | 0.066 | 0.373 | 0.873 | | | | | | |
| DX | 0.109 | 0.401 | 0.232 | 0.860 | | | | | |
| DA | 0.217 | 0.821 | 0.380 | 0.415 | 0.879 | | | | |
| DC | 0.501 | 0.673 | 0.246 | 0.314 | 0.572 | 0.926 | | | |
| DO | 0.365 | 0.638 | 0.402 | 0.455 | 0.785 | 0.472 | 0.833 | | |
| DI | 0.206 | 0.714 | 0.616 | 0.733 | 0.720 | 0.576 | 0.729 | 0.843 | |
| DR | 0.241 | 0.675 | 0.469 | 0.501 | 0.791 | 0.637 | 0.805 | 0.801 | 0.807 |

Source: Survey data processing by the authors.

*4.2. Model Fitting*

The fitting effect of the structural equation model indicates whether the interaction between variables exists. By testing the fitting effect of the model, the model is continuously optimized until the model with the best fitting effect is found. The used goodness-of-fit indices of the structural equation model are: ratio of chi-square to degree of freedom ($\frac{x^2}{df}$); goodness-of-fit index (GFI); standardized root mean square residual (SRMR); root mean square error of approximation (RMSEA); normed fit index (NFI); incremental fit index (IFI); and comparative fit index (CFI).

After testing the model, it was found that it was necessary to exclude the critical infrastructure security factor (SI5) with the value of 0.41, because it did not meet the ideal standard of 0.5 [122].

The Modification Indices (MI) option in SPSS Amos 26.0 software is used to find the path with the maximum MI value and to add the path, that is, the initial model is modified once; then, the SPSS Amos 26.0 software is run to obtain the goodness-of-fit index value of the modified model to verify the fitting effect of the modified model. The above steps are repeated until the value of the goodness-of-fit index of the modified model approaches or meets the requirements of the standard value. According to the above method, the initial model was modified several times, and finally the optimal structural equation model was obtained, as shown in Figure 2.

The values of the goodness-of-fit indices of the modified structural equation model are shown in Table 6.

**Table 6.** Fit summary of criteria and modified model.

| Fit Indices | $\frac{x^2}{df}$ | GFI | SRMR | RMSEA | NFI | IFI | CFI |
|---|---|---|---|---|---|---|---|
| Recommended value | <2 | >0.90 | <0.08 | <0.06 | >0.90 | >0.90 | >0.90 |
| Source | [123] | [124] | [125] | [125] | [126] | [126] | [127] |
| Modified model | 1.412 | 0.913 | 0.076 | 0.057 | 0.904 | 0.966 | 0.959 |

Source: Survey data processing by the authors.

Most of the fit indices of the modified model were improved. The fit degree of the model is relatively good and basically passes the goodness-of-fit test. In other words, the correlation between variables in the modified structural equation model shown in Figure 2 does exist.

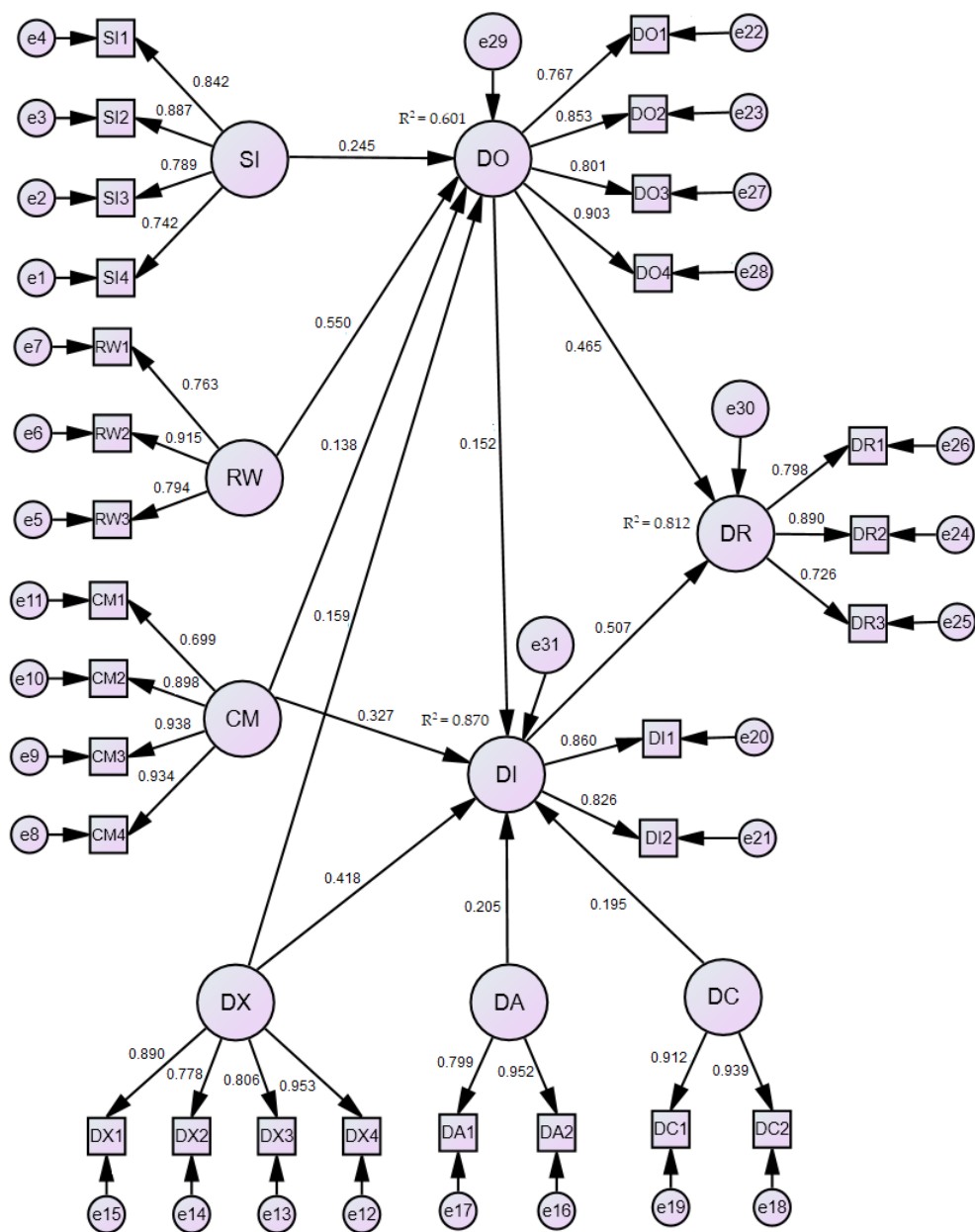

**Figure 2.** SEM measurement model. Source: Developed using SPSS Amos v.26.0.

### 4.3. Analysis of Direct and Mediation Effects

Figure 2 shows that for investments in digital transformation, the value of $R^2$ is 0.601, which means 60.1% of the variance; therefore, the predictive power of this construct is strong. At the same time, the values of $R^2$ for digital innovation and digital resilience are 0.870 (87% of variance) and 0.812 (81.2% of variance), respectively, which means that the predictive power of these constructs is very strong.

The path coefficient and hypothesis testing results of the theoretical model are shown in Table 7.

**Table 7.** Hypothesis testing results.

| Hypothesis | Paths Correlation | Path Coefficient (β) | p | Results |
|---|---|---|---|---|
| H1 | Security investments → Digital core | 0.245 | *** | Supported |
| H2 | Remote working → Digital core | 0.550 | *** | Supported |
| H3 | Cloud migration → Digital core | 0.138 | *** | Supported |
| H4 | Digital transformation investments → Digital core | 0.159 | *** | Supported |
| H5 | Cloud migration → Digital innovation | 0.327 | *** | Supported |
| H6 | Digital transformation investments → Digital innovation | 0.418 | *** | Supported |
| H7 | Digital adaptation → Digital innovation | 0.205 | *** | Supported |
| H8 | Digital acceleration → Digital innovation | 0.195 | *** | Supported |
| H9 | Digital core → Digital innovation | 0.152 | *** | Supported |
| H10 | Digital core → Digital resilience | 0.465 | *** | Supported |
| H11 | Digital innovation → Digital resilience | 0.507 | *** | Supported |

Note: *** $p < 0.05$. Source: Survey data processing by the authors.

According to the data in Table 7, the $p$ values of the expected hypotheses H1–H11 are all less than 0.05, indicating that these hypotheses are supported by the data.

Thus, we analyzed the direct effects between DO and its predictors (SI, RW, CM, and DX), DO and its successors DI and DR, DI and its predictors (CM, DX, DA, and DC), and DI and its successor DR.

From the direct effects analysis, the following results were obtained:

- RW has a positive and significant impact on the DO (β = 0.550; $p < 0.05$), which indicates the importance of remote work in terms of core digital investments;
- The impact of SI on DO (β = 0.245; $p < 0.05$) is positive and statistically significant, which supports the positive role of digital security in the ability of companies to make core digital investments;
- DX has a significant and positive impact on both DO (β = 0.159; $p < 0.05$), which suggests that the entire suite of IT solutions and applications has an important role in terms of the basic capacity of the digital core investments, and on DI (β = 0.418; $p < 0.05$), which demonstrates the role of the technologies and applications used by the company on digital innovation;
- CM has a significant and direct impact on both DO (β = 0.138; $p < 0.05$), which means that migration to cloud technologies is a decisive factor in terms of core digital investment, and on DI (β = 0.327; $p < 0.05$), which highlights the role of cloud computing technologies and in the digital innovation capacity of companies;
- The impact of DA on DI (β = 0.205; $p < 0.05$) is positive and statistically significant, which supports the importance of IT support solutions for vulnerable areas caused by pandemics and crises or for the new requirements of the operating system regarding digital innovation at the company level;
- DC has a positive and significant impact on DI, close to that of DA (β = 0.195; $p < 0.05$). This result indicates the importance of IT solutions that change the business process or allow increasing the market share in terms of the digital innovation capacity of companies;
- The positive and significant coefficient between DO and DI (β = 0.152; $p < 0.05$) means that the core digital investments at the level of companies produce a positive and significant impact on their digital innovation;
- DI has a positive and significant impact on DR (β = 0.507; $p < 0.05$) and DO has a positive and significant impact on DR (β = 0.465; $p < 0.05$), respectively. These results indicate the importance of both digital innovation and core digital investment in the digital resilience of companies in the context of the pandemic and the crisis.

The high values of the coefficients of the endogenous latent variable (DR), "DR1", "DR2", and "DR3", respectively, mean that the digital resilience of companies is well represented by these indicators.

In addition to the analyzed direct relationships, in the proposed model, we identified and analyzed the mediating effect of DO on the relationship between SI, RW, CM, DX, and DI, respectively, and the mediating effect of DI on the relationship between DO and DR.

Table 8 shows the specific standardized total, direct, and indirect effects; the direct ones are in accordance with Table 7, and the $p$ values of the structural relations are all less than 0.05.

**Table 8.** The standardized total, direct, and indirect effects registered between the variables of the structural equations modeling.

|  | SI | RW | CM | DX | DA | DC | DO | DI |
|---|---|---|---|---|---|---|---|---|
| | | | | Standardized Total Effects | | | | |
| DO | 0.245 | 0.550 | 0.138 | 0.159 | 0.000 | 0.000 | 0.000 | 0.000 |
| DI | 0.037 | 0.084 | 0.348 | 0.442 | 0.205 | 0.195 | 0.152 | 0.000 |
| DR | 0.133 | 0.298 | 0.240 | 0.298 | 0.104 | 0.099 | 0.542 | 0.507 |
| | | | | Standardized Direct Effects | | | | |
| DO | 0.245 | 0.550 | 0.138 | 0.159 | 0.000 | 0.000 | 0.000 | 0.000 |
| DI | 0.000 | 0.000 | 0.327 | 0.418 | 0.205 | 0.195 | 0.152 | 0.000 |
| DR | 0.000 | 0.000 | 0.000 | 0.000 | 0.000 | 0.000 | 0.465 | 0.507 |
| | | | | Standardized Indirect Effects | | | | |
| DI | 0.037 | 0.084 | 0.021 | 0.024 | 0.000 | 0.000 | 0.000 | 0.000 |
| DR | 0.133 | 0.298 | 0.240 | 0.298 | 0.104 | 0.099 | 0.077 | 0.000 |

Source: Survey data processing by the authors.

From Table 8, we observed the following indirect effects due to mediation:

- The mediation effect of DO on DI through the four exogenous latent variables SI (β = 0.037; $p < 0.05$), RW (β = 0.084; $p < 0.05$), CM (β = 0.021; $p < 0.05$), and DX (β = 0.024; $p < 0.05$) has an indirect positive impact on DI, which contributes to the intensification of the total effect exerted by CM (β = 0.488; $p < 0.05$) and DX (β = 0.442; $p < 0.05$) on DI;
- DI mediates the effect between DO and DR as a result of the indirect positive effect of the seven latent variables on DR (SI with β = 0.133, $p < 0.05$; RW with β = 0.298, $p < 0.05$; CM with β = 0.240, $p < 0.05$; DX with β = 0.298, $p < 0.05$; DA with β = 0.104, $p < 0.05$; DC with β = 0.099, $p < 0.05$; and DO with β = 0.077, $p < 0.05$).

As a result of the indirect effect exerted by DO on DR through the mediating effect of DI, it was found that the most significant total positive effect is that of DO (β = 0.542; $p < 0.05$) on DR, followed very closely by the significance of DI (β = 0.507; $p < 0.05$), though this comes only from the direct effect.

Therefore, empirical data confirm the H12 hypothesis.

In the interpretation of SEM results, the analysis of mediation effects is of particular importance. Thus, in the model optimized for direct effects, it was found that DI has a positive and strong impact on DR, followed very closely by DO, and in the case of total effects DO has a positive and strong impact on DR, followed very closely by DI.

## 5. Conclusions

In the technological revolution 4.0, digital technologies affect all companies' activities, forcing a change in the way to do business. At the same time, the global COVID-19 crisis has made companies more aware of the need to implement 4.0 technology, which is essential for the resilience of modern businesses [3]. Much research has addressed how companies approach crisis and disaster recovery [11–15], but it is very important to focus on the need to build digital resilience to ensure business resilience. Organizations must not only respond quickly to threats, but also learn to opportunistically rise above them. The digital world calls for a new technology-based approach to coping with future crises—digital resilience.

The main objective of this study was to address a significant issue regarding the impact of the digital transformation of companies in order to ensure digital resilience. In this regard, based on the specialized literature, we have developed a conceptual model regarding digital resilience on the example of Romanian companies, based on the important components of digital core and innovation investments. Thus, we found that there are direct causal relationships between digital core investment and its components; digital innovation and its components; and between digital innovation, digital core, and digital resilience. We also identified two indirect causal relationships through a mediating effect: (1) the digital core effect on the relationship between its direct factors and digital innovation and (2) the digital innovation effect on the relationship between digital core and digital resilience. At the same time, the analysis of direct and indirect effects demonstrated the positive and significant impact of digital core and digital innovation on digital resilience, but in the reverse order of factors (digital innovation has the strongest direct impact and digital core the strongest indirect impact).

### 5.1. Theoretical Implications

This research deepens the theoretical understanding of how the technological revolution 4.0 influences the digital transformation of companies. Thus, by analyzing the relationships between digital transformation, innovation, and digital resilience, the study provides relevant guidance for businesses to obtain resilience through digital transformation.

Starting from the economic modeling to determine the influence of digital technologies on the activity of companies and using structural equations modeling, this study identifies the main factors that can ensure digital resilience and assesses their impact on private companies and public institutions from Romania.

The results show that all 12 hypotheses of the structural model are valid, being consistent with the results of other specialized studies. According to the model, the four variables of core digital investments (security investments, remote working, cloud migration, and digital transformation investments) have a positive and significant effect on the digital core. The analysis of direct results indicates the importance of remote work (RW) [46,49,50,83] and supports the positive role of digital security (SI) in the ability of companies to make core digital investments [44,45]. At the same time, the use of IT solutions and applications (DX) and migration to cloud technologies (CM) [57–59] have an important role to play in increasing digital core capacity (DO) [81].

It was found that the impact of digital adaptation (DA) and digital acceleration (DC) on digital innovation (DI) is positive and statistically significant, which supports the importance of IT solutions in conditions of vulnerability generated by the pandemic and crisis [70,80], solutions that change the business process at the company level, and adapting to new requirements based on digital innovation. At the same time, the results identify the significant role of the technologies and applications used by the companies (DX) and the migration to cloud technologies (CM) in their digital innovation capacity (DI) [11,72,101].

According to the model, it can be observed that digital core investments (DO) at the company level have a significant positive impact on digital innovation (DI) [22,103] and their digital resilience (DR). Moreover, the analysis of direct results shows that the digital resilience (DR) of companies is well represented by digital innovation (DI) and digital core investments (DO), which have a direct, positive, and strong impact on ensuring the digital resilience (DR) of companies in the context generated by the pandemic.

The indirect effects analysis highlights the role played by the two mediating variables: digital core investments (DO) and digital innovation (DI), which intensify the result determined by the analysis of the direct effects. Thus, the mediating variable digital core investments (DO) amplifies the impact on digital innovation (DI) through the four independent variables of remote working (RW), security investments (SI), digital transformation investments (DX), and cloud migration (CM) (the last two having the strongest direct effect on digital innovation (DI)). The mediation variable digital innovation (DI) amplifies the

impact on digital resilience (DR) through the seven latent variables (RW, DX, CM, SI, DA, DC, and DO).

The results obtained through the analysis of the two types of direct and indirect effects demonstrate that, in the case of the direct effects, digital innovation (DI) has a significantly more powerful impact than digital core investments (DO) on digital resilience (DR), while, considering the indirect effects, DO has a more powerful impact than DI on DR, though the values are close.

*The novelty that could contribute to the enrichment of the literature is given firstly by the integration of variables specific to digital technologies and innovations in resilience in the framework of the developed structural model. Secondly, through the mediation analysis, a better understanding of the relationships between the independent and dependent variables is achieved by adding a dimension that completes the causal relationships determined by the direct effects.*

### 5.2. Managerial Implications

This study offers several managerial implications that company managers can consider to ensure digital resilience. Thus, organizations must accelerate the pace of implementation of digital technologies and the adoption of digital innovation to ensure resilience to uncertainty and proactive change to withstand any crisis type. This study helps to identify strategies and implementation plans that allow a rapid reaction to market changes and thus increase the degree of resilience on several levels:

1. Adaptive digital innovation-based management is needed to provide extensive digital communication skills;
2. The digital capacity of employees should be improved and they should be involved in the process of digitizing the company;
3. The widespread use of software applications should be implemented both for the analysis and interpretation of data in order to substantiate and make decisions in real-time, and to improve the relationship with customers and increase the quality of services and products offered;
4. An organizational culture based on digital technologies should be implemented to allow faster and more efficient responses in the face of a crisis;
5. IT procedures and applications should be integrated to detect shocks promptly and assess their impact as they occur;
6. It is necessary for companies to implement flexible measures for allocating financial resources, with priority to digital investments, to ensure the stability and development of the business.

The evaluation of a company, depending on these aspects, can also be conducted during and after the crisis, in order to identify vulnerable areas and plan actions to reduce or eliminate them. Assessing a company's resilience depends on the specifics and context of each crisis. The two stages, recovery from shock and preparation for future shocks, must be linked.

The results of the study could benefit business managers who, by integrating these variables, will be able to develop new products and services based on digital innovation, which can lead to increased economic performance and building digital resilience. Given the results of the research, a proposal for the managers of the organizations could be to make sound planning and management of the budgets related to ensuring digital resilience. Romanian companies, as they strive to emerge from the pandemic crisis as well as the one caused by the Russo–Ukrainian war, have the opportunity to shape the economic recovery in a way that offers sustainable change and a long-term impact. Managers who understand (and can capitalize on) what worked well in downtime will be best positioned to experience growth and overcome competition once normalcy is restored.

The practical contribution of this study is to understand the problems faced by Romanian companies in the conditions of pandemic, crisis, or war, correlated with the need to

implement digital technologies and innovations to ensure digital resilience and support future business.

*5.3. Limitations and Future Research Directions*

This study has some limitations which suggest future research directions. First of all, this study only concerns organizations operating in Romania, which limits the generalization of the results obtained because the specific characteristics of the country can influence how digital transformation and innovation impact digital resilience and the size of this impact. Future research should therefore conduct cross-country comparative analyses to verify that the findings of this study are valid for other countries as well.

Another limitation is determined by the type of information taken from the questionnaire (it contained only closed questions) and the types of scales being reduced in variety.

Another limitation is that respondents may, when completing the questionnaire, show subjective bias or be reluctant to provide information.

Future research is intended to remove these limitations by combining the use of quantitative and qualitative methods, which may lead to the identification of other factors at the level of companies that contribute to ensuring digital resilience. Potential future research directions could extend the analysis conducted in this study by using a quantitative and qualitative comparative analysis on clusters.

**Author Contributions:** Conceptualization, A.M. and G.S.; formal analysis, A.M. and G.S.; investigation, A.M. and G.S.; methodology, A.M. and G.S.; resources, A.M. and G.S.; software, A.M. and G.S.; validation, A.M. and G.S.; writing—original draft, A.M. and G.S. All authors have read and agreed to the published version of the manuscript.

**Funding:** This research received no external funding.

**Acknowledgments:** The authors acknowledge the anonymous reviewers whose suggestions and comments helped improve the paper.

**Conflicts of Interest:** The authors declare no conflict of interest.

## Appendix A

**Table A1.** Survey content.

| 1. | Gender | Female<br>Male |
|----|--------|----------------|
| 2. | Age | <25<br>25–30<br>31–40<br>41–50<br>>50 |
| 3. | Education | College diploma<br>Bachelor's degree<br>Master's degree<br>Ph.D. degree and above |
| 4. | Type of enterprise | Small/Private enterprise<br>Medium/Private enterprise<br>Large/Private enterprise<br>Public/State-owned enterprise |
| 5. | IT budgets (IT) | <5% of total revenue<br>5–10% of total revenue<br>11–20% of total revenue<br>>20% of total revenue |

**Table A1.** *Cont.*

| | | |
|---|---|---|
| 6. | Enterprise's age | <5<br>5–10<br>11–20<br>>20 |
| 7. | Enterprise location | Urban<br>Rural |
| 8. | Security investments | The company ensures Internet of Things security<br>The company uses solutions application security<br>The company ensures network security<br>The company uses solutions cloud security<br>The company uses solutions for critical infrastructure security |
| 9. | Remote working | The company allows fully remote work<br>The company allows hybrid remote work<br>The company does not allow remote work |
| 10. | Cloud migration | The company uses software as a service (SaaS)<br>The company uses platform as a service (PaaS)<br>The company uses infrastructure as a service (IaaS)<br>The company uses a hybrid model |
| 11. | Digital transformation investments | The company uses tools for interaction through the website, e-commerce, and m-commerce<br>The company uses mobile applications in business processes—Apps<br>The company uses social networks in the digital transformation of business—Social media<br>The company uses conversational interfaces—Chatbots |
| 12. | Digital adaptation | The company implements IT projects to support vulnerable processes discovered in times of crisis<br>The company develops IT projects in support of the new operational requirements generated by the implementation of technology 4.0/pandemic crisis |
| 13. | Digital acceleration | The company develops IT projects that model business innovation<br>The company develops IT projects to increase market share |
| 14. | Digital core | The company adopts industry 4.0 technologies in the automation process<br>To what extent has the company achieved digitizing the business<br>To what extent the digitalization of the business has led to its globalization |
| 15. | Digital innovation | The company introduces digital products<br>The company uses customer touch points and gives enhanced sales pitches |
| 16. | Digital resilience | The company has established a strategy for developing the digital business model<br>The company has improved customer experience: customer journeys, channels, and touchpoints<br>The company uses platforms and infrastructure for digital processing of the data and information |

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
