# Peer review of "A Structural Framework for Assessing the Digital Resilience of Enterprises in the Context of the Technological Revolution 4.0"

_electronics, doi:10.3390/electronics11152439_

Round 1
Reviewer 1 Report
The topic is very interesting and will add to the state of the art.
I recommend publication with minor revisions. There needs to be some justification of the methodology used in the first 2-3 paragraphs of the methods section by citation.
The abstract is clearly articulated, the introduction and literature background is very thorough.
The results are clearly discussed and very excellent statistical analysis is carried out.
In summary very well written and interesting area of research.
Author Response
August 01, 2022
The authors are grateful to the Reviewer of manuscript number electronics-1832220 for his thoughtful comments.
Response to Reviewer 1 Comments
The comments of the reviewer are taken into consideration and answered in this letter.
The authors believe that the revised version of the paper is better than the originally submitted one. For this reason, they reiterate their gratitude to the anonymous reviewer.
​In paper revision we used the "Track Changes" function in Microsoft Word.
We would like to thank the Reviewer for kind evaluation of the paper and the positive feedback of our work. The entire article has been revised carefully.
“The topic is very interesting and will add to the state of the art.
I recommend publication with minor revisions. There needs to be some justification of the methodology used in the first 2-3 paragraphs of the methods section by citation.
The abstract is clearly articulated, the introduction and literature background is very thorough.
The results are clearly discussed and very excellent statistical analysis is carried out.
In summary very well written and interesting area of research.”
- Authors: We thank the reviewer for your appreciation and the suggestion. Please see the modified Methodology, section 3.2. Structural Equation Modeling (SEM).
Thank you for your thoughtful review. We believe we have responded satisfactorily to your concerns.

Reviewer 2 Report
This research establish the influence of digital core investment and digital innovation on digital resilience at the enterprise level which is useful for enterprises. The data sources are reliable and the methods are suitable to answer the research questions. The results are acceptable. Two main questions need to be revised. 1) The data sources of Table.1 need to be clarified at Table.2 at Line 372 in Page 10. 2) The results analysis needs to be rewritten to explain the effects in detail.
Author Response
The comments of the reviewer are taken into consideration and answered in attached letter.
The authors believe that the revised version of the paper is better than the originally submitted one. For this reason, they reiterate their gratitude to the anonymous reviewer.

Reviewer 3 Report
Thank you for the opportunity to read the paper. It is an interesting toping and I consider it fits to the journal. The article reports on a very interesting study that may reach large audiences. The article is very well organized, in conceptual and methodological terms, and presents very relevant results. The research question is clearly stated. The theoretical framework is creative. The research question is explored in a way that is new, creative and important to the discipline. The methodology is clearly explained. The empirical data are analysed in appropriate ways, and written up in ways that are easy to understand. The study conclusions supported are by the analysis. The biography is rich and up-to-date. The authors have done an excellent job.
Author Response
We would like to thank the Reviewer for kind evaluation of the paper.
The authors reiterate their gratitude to the anonymous reviewer.
Round 2
Reviewer 2 Report
The data source has been revised clearly and readers would understand the data sources now. Since the variables has been revised in table 1, data summarization of all variables in Table 1 need to be supplemented in Table 2. Moreover, the direct and indirect effects of conclusions need to be rewritten clearly. It's important to explain the reason and mechanism that how to form such effects at theory implications.
Author Response
August 02, 2022
Response to Reviewer 2
The comments of the reviewer are taken into consideration and answered in this letter.
The authors believe that the revised version of the paper is better than the originally submitted one. For this reason, they reiterate their gratitude to the anonymous reviewer.
​In paper revision we used the "Track Changes" function in Microsoft Word.
We would like to thank the Reviewer for evaluation of the paper.
“The data source has been revised clearly and readers would understand the data sources now. Since the variables has been revised in table 1, data summarization of all variables in Table 1 need to be supplemented in Table 2. Moreover, the direct and indirect effects of conclusions need to be rewritten clearly. It's important to explain the reason and mechanism that how to form such effects at theory implications.“
- Authors:
We thank the reviewer for the observation and suggestion.
So, we added Appendix A Table A1 with data summarization of all variables.
We rewritten conclusions with the direct and indirect effects and we explained them in lines 639-678.
Thank you for your thoughtful review. We believe we have responded satisfactorily to your concerns.